# The Skeletal Remains of Soldiers from the Two World Wars: Between Identification, Health Research and Memorial Issues

**DOI:** 10.3390/genes13101852

**Published:** 2022-10-13

**Authors:** Marine Meucci, Emeline Verna, Caroline Costedoat

**Affiliations:** 1Centre National de la Recherche Scientifique, Établissement Français du Sang, ADES Laboratory Unité Mixte de Recherche 7268, Aix Marseille University, 13344 Marseille, France; 2National Office for Veterans and Victims of War (ONACVG), 75007 Paris, France

**Keywords:** biological anthropology, biological profile, DNA analysis, genetic and social identification, health crisis, missing person, wars, NGS, skeletal remains

## Abstract

After causing mass disasters that claimed the lives of tens of thousands of soldiers from countries around the world, the two Great Wars left some of them lost and missing. In France, these corpses reside in a legal vagueness where they belong neither to forensic anthropology nor archeology. Nevertheless, the process of identification and determining the cause of death requires the use of modern forensic anthropology by applying biological profiling and DNA analysis. New genomic methods also provide insight into the health statuses of these military populations, providing new perspectives on these periods of humanitarian crisis.

## 1. Introduction

Humanitarian disasters are now, more than ever, a challenge for forensic sciences in terms of the search, identification and gestion of remains. The two World Wars are among the worst humanitarian crises, having claimed the lives of more than one million soldiers. The mass deaths from these periods left thousands of bodies on the battlefields and led to problems that are still prevalent today. For the first war (1914–1918), the French government initiated a campaign to search for and return bodies to decent graves [1]. More than 700,000 soldiers are still missing today, but the expansion of the modern landscape with construction work turns the soil and brings back memories of those times of war. Every year, the Office National des Anciens Combattants et Victimes de Guerre (ONACVG—National Office of War Veteran and War Victims) documents tens of bodies of soldiers who have been missing for more than 100 years. These bodies are covered by law with the phrase “Mort pour la France” (“Died for France”), and by right, they can return to their families and be buried in the family vault or be buried with their brothers in arms in a military necropolis. The identification process for these soldiers is important, as much for memorial duty as for resolving missing cases for grieving families that have been unresolved for 100 years, due to the war catastrophes. In France, deaths for both Great Wars (1914–1918 and 1939–1945) are not recognized as forensic cases, even regarding the nature of modern mass disasters and the use of tools of forensic anthropology and DNA to recover identities, as was the case for the discovery of numerous bodies of soldiers from the Commonwealth in Fromelles, for example [2] Any evidence, whether biological or artefactual, is part of the identification process [3]. A whole investigation is carried out based on this double process of identification.

This Special Issue on the “Identification of Human Remains for Forensic and Humanitarian Purposes: From Molecular to Physical Methods” is an opportunity to present the conceptual, methodological and technical advances made by various institutions in France in the context of recent conflicts.

Describing and understanding what kind of informal social and biological profiles can contribute to this identification process is important for studying humanitarian crises and its necessary interdisciplinarity. The study presented here was based on more than 10 years of experience and reports on the results and methods used in the identification of more than 150 soldiers of the First and Second World Wars.

## 2. Identification Process: Cross-Referencing Social and Biological Data

To determine identity, several methods are applied to achieve a high degree of accuracy and to narrow the focus of the search to the individual identification of these soldiers. Biological profiling involves the estimation of sex, age and stature and describes the pathologies and traumatic patterns. Personal effects and/or military artefacts can also be used to retrieve the identity of a missing soldier. This chapter is a review of almost 10 years of searching, with 153 skeletons excavated and studied, 23 of which were successfully identified. For soldiers from both World Wars having a particular juridical framework, methods from biological anthropology (forensic and archeological context) are used, even though this process belongs to none of these research fields. We will look at these criteria in more detail in the following sections.

### 2.1. Biological Profile

#### 2.1.1. Sex

Even though the military context in 1914 to 1945 was mostly represented by men (particularly in the European armies), sex estimation is necessary for accurate biological profiling. Pelvic bones are the most dimorphic bones in the human body, as their shape is influenced by hormones during puberty [4]. Biological differences between males and females, such as giving birth, make pelvic dimorphism the best marker for sex estimation [5]. In France, both morphometric [6] and morphoscopic [7] methods are used directly on the coxal bones according to Bruzek’s method, with 95% accuracy (e.g., Table 1). Sex estimation in such populations where the sex is already known from the context is an opportunity to assess the method’s accuracy and could generate new ideas for developing more efficient methods, especially in combination with morphoscopic methods.

Even though this is not performed on a regular basis, DNA can also be used to verify/validate the sex of the individual if necessary. Current techniques allow for the detection of sex using different methodologies depending on the quantity and quality of the DNA present. In France, the preferential method is measuring amelogenin. In brief, the length of the gene coding for amelogenin varies by 6 bp according to whether it is on chromosome X or Y. By measuring this gene’s length, it is possible to accurately determine sex, with up to 100% accuracy [8,9,10,11].

#### 2.1.2. Age

The skeletal age at death is a challenge for biological anthropology, as it is calculated using the degree of senescence observed on different parts of the skeleton, which could vary according to interindividual variability [3,12,13,14,15]. For the identification process, the age at death needs to be determined as accurately as possible. Regarding the military context, some individuals were very young (in France, the legal age to enroll was 18 years in 1914), and epiphysial fusion may not yet have been finished. Therefore, several methods are used, one of them especially for young individuals (e.g., Table 2) [16]. Pelvic bones, as for sex estimation, are the preferential bones for determining the age at death: the auricular surface and pubic symphysis are the most used in forensic anthropology [12,17]. Methods applied to the entire body also need to be underlined in such contexts: regarding mass disasters from wars, it is useful to determine the MNI (minimum number of individuals), as bodies become mixed during catastrophes and emergencies. In seeking to individualize bodies, looking for age markers applicable to the entire body [18] could help with the association of bones with some others based on determining whether the skeleton is generally youthful or otherwise.

As an extreme example, a body from a 21-year-old man will not have the same epiphyseal fusion and degenerative markers as a 40-year-old man. Skeletal age estimation is not sufficiently accurate for use as the only marker of identification; however, different methods help to effectively discriminate between a young adult, a mature adult and an old adult, each category is based on the levels of degenerative markers [19]. As we can observe in Table 3, the skeletal age estimation for a 20-year-old soldier provided a correct range for the age but was not accurate enough to be a discriminating criterion for identification, especially in contexts where several victims can be within the same age ranges. Using multiple methods of age estimation can help to more accurately determine the skeletal age.

Regardless of the methods used, estimating the age at death is difficult, and the margin of error is often large, especially for adults. Age estimation for adults is therefore a real challenge, and, although the genetic tool has not yet proved its worth in addressing these problems, the prospects offered by the various methods for refining the age ranges of adults are promising.

Epigenetic data recovered from so-called clock-CpGs coupled with statistical models or even machine-learning approaches offer the possibility of more precisely estimating the age distributions at the death of individuals, with an accuracy of 4.5 years in some cases [20,21,22]. Their use should advantageously refine current estimates based on developmental growth models (of limited value for adult individuals) and will provide a basis for new study perspectives. Calibrations still need to be conducted according to the tissues used or the populations studied, but these new genomic methods offer the possibility of obtaining demographic data on different populations; thus, expanding the panel of reference populations in a forensic context is essential.

#### 2.1.3. Stature

To evaluate the stature of a soldier, the preferential method is based on long bone measurement [23]. When possible, the bones used are the humerus, femur, and tibia because they provide the highest accuracy regarding Olivier’s degree of trustworthiness (0.889, with a margin of error of +/−3.05 cm). Stature determination in a military context, as is the case for modern missing cases, is important information to compare with official documents (Figure 1). As is the case for age, stature determination can help in individualization, as a man measuring 1.57 m will not have the same bone length as a man measuring 1.84 m.

#### 2.1.4. Pathologies

In the search for identity, every clue leading to the singularity of a body compared to another is precious information. As for stature and age, the military documents mentioned some medical particularities for each soldier (e.g., Figure 2). Unfortunately, tooth conditions are rarely mentioned, except for very specific ones such as dentures. However, condition indications are still useful for distinguishing one identity from another. The pathologies sought with priority are antemortem fractures (with a consolidation long before the war), amputations, any noteworthy features on bone shapes, or dentures. Anatomical variations are also important for such identity-search purposes and can also be used in forensic anthropology [24]: those with visible body consequences, such as spina bifida occulta, can also be reported in military documents.

### 2.2. Social Identification

Social identification is defined here as the possibility of retrieving an identity by any means other than biological and genetic profiling. For mass disasters such as wars, this type of identification can especially focus on personal effects and archives.

#### 2.2.1. Personal Effects

Besides biological profiling, a large part of identity research is based on artefacts found with corpses. According to their nature, they provide different information (e.g., Table 4). Some objects can provide temporality, such as the famous red French pant from the beginning of WWI, while others can indicate nationality, and some can even provide identities (e.g., Figure 3). Given the known importance of artefacts, fine excavation, as performed in archeology, is essential. The association of an artefact with the right skeleton, determining its position and, even more simply, finding it are key steps in the identification process. Through the years, 19 soldiers have been identified thanks to artefacts, mostly by military ID tags but also by engraved personal objects.

#### 2.2.2. Archivistic Search

French missing soldiers classified as “Mort pour la France” (Death for France) are listed on the government website Mémoire des Hommes (Human’s memory) [25]. Every dead soldier from both World Wars can be found and identified if the skeleton and the artefact allow: it is possible to have the name and information of every man who died during the First and Second World Wars. The different information one can fill in include the name, last name, date of birth, date of death, regiment and grade. It is interesting to underline that a search can also be based on the place of death, which, regarding today’s discovery context, could be useful. After filling in the different information, a list of individuals is provided. If his exact name is not known, it will be necessary to check every military detail listed to see if other information in possession can fit, such as regiment. Consulting the Death for France form provides the places of birth for all soldiers, allowing the search of the register form only available in departmental archives (e.g., Figure 1). For each soldier name, a comparison between the biological profile and information in the register can be made, but using only the stature or age correspondence does not provide good accuracy.

When the name is known, the departmental register provides the parent’s name, which is the first step for a genealogic search. The National Office of War Veterans and War Victims has a specific department in charge of family searches; they are able to contact the town hall to acquire a historical overview of the family and allow DNA analysis if necessary.

### 2.3. DNA Identification: Comparison of Genetic Profiles

The final, non-mandatory step is the identification of the victim by comparing the DNA profiles with those of family members. Although the biological profile and the artefact may provide sufficient information for determining an identity, DNA analysis by DNA profile comparison remains the most accurate means of validating a hypothetical identity and, above all, providing a reliability value for that identity (the gold standard).

The differential study of the two types of DNA (mitochondrial or nuclear) can provide maternal information (upon studying the mitochondrial DNA, e.g., Figure 4); paternal information (upon studying the Y chromosome); or information on both parents in the case of autosomes, since each child has a maternal copy and a paternal copy. The degree of relatedness determines how much of the gene pool two individuals will have in common [26]. Parents and children share half of their DNA, and an individual will have, on average, a quarter of the DNA of their grandparents and grandchildren. Unless mitochondrial DNA or Y-chromosome analysis is carried out, the most reliable comparisons will be made using samples taken from the parents or children of the missing persons.

Advanced research in DNA, especially in the disaster context, seeks progress in accessibility, accuracy and speed. The most common method used is genetic identification by STR profile comparison: STR (short tandem repeat) markers are amplified and compared to one another [8]. Regarding STR marker choice, chromosome-X-STR markers tend to be more stable than Y-STR and provide better quality for identification [27]. As DNA tends to degrade with time [28], simple DNA comparisons are not always feasible: Y-STR analysis (Y chromosome, paternal lineage) and the analysis of other nuclear DNA cannot be performed in this case [3]. Other studies show, however, that mitochondrial DNA (maternal lineage) is preferential for analysis in such a context [29,30,31].

DNA from bones (long bones or teeth) is amplified, sequenced [32] and compared to the DNA of buccal cells from a living parent. This part of the process becomes complicated, especially for those who went missing in WW1. As time goes by, and generations follow one another, it becomes more and more difficult to find a living person whose DNA comparison would allow a certain identification of the soldier found [26].

In addition to the differential properties of the different markers, which make it possible to work even under extreme conditions of DNA preservation, next-generation sequencing has also enabled new molecular strategies for genetic identification [32,33,34].

If DNA verification is requested for a soldier’s identity, then a comparison is made between the DNA profile from the bone remains and the DNA of the family member (s) who have agreed to be linked. To date, 4 out of 153 soldiers have been studied by DNA comparison.

## 3. Characterization of Trauma Patterns and Health Conditions

The two World Wars show similarities in their mass deaths due to modern military equipment. Especially in the First World War and trenches, the conditions of life tended to deteriorate with humidity, rats, fleas, lice, sludge, dirt and cold. Therefore, soldiers died not only from violence but also from infections and diseases. Even though biological anthropology cannot ascertain with certitude one’s cause of death, a large bundle of evidence can help in appreciating the sanitary context during the war and its effect on mortality. The observation and identification of lesions allow hypotheses on violence, wounds and causes of death.

### 3.1. Trauma and Violence Testimony

As surprising as it may seem, historical sources provide very little information on the real causes of death in combat. On such a subject, it is obvious that a paleopathological reading of the remains of soldiers constitutes an exceptional opportunity to objectify, in many cases, peri-mortem traumas, some of which may have been the causes of death (e.g., Figure 5). No body parts are exempt from trauma, and ballistic impacts are observed on the skull as well as the rest of the skeleton.

Even though ballistic trauma is the most common, trauma analysis in the war context can reveal traumas described in forensic anthropology [35]: sharp force trauma, thermal trauma, firearm trauma and blunt force trauma. Diseases have always played a role in most soldiers’ deaths, but with the appearance of modern arms, more individuals die from wounds. Innovation in terms of arms left numerous men with heavy injuries: for example, shrapnel killed many but injured as many, causing greater logistic mobilization in terms of medical support [36,37]. Some traumas such as blast effects can explain certain discovery contexts such as those buried by a large quantity of soil after a shell explosion, for example.

Trauma analysis is a good window into the reality of war, its violence and its cruelty. Bodies are bruised and battered, with wounds. Analyzing these wounds contributes to the individual’s story and the duty of remembrance. Moreover, war contexts allow the observation of the same injuries and traumas; some weapons and death strategies do not necessarily evolve or change. This kind of information can be useful for a better picture.

### 3.2. Diseases: The Hidden Causes of War Deaths

With the difficult living conditions in wars and context of great inter-individual proximity, epidemics are more likely to appear and spread, particularly those of infectious origin [38]. Numerous analyses, including paleoserology, have shed light on certain pathogens that occur in the context of war [39]. One example is trench fever, whose pathogen, *Bartonella quintana*, is transmitted by body lice and causes disabling symptoms on a battlefield, particularly because they recur over several weeks [40,41]. Trench fever caused nearly 150,000 deaths among French soldiers between 1914 and 1918 [42]. Trench fever is promoted by proximity and poor living conditions; the bacterium is still present today and remains a challenge in refugee camps, for example; body lice management and eradication are still necessary [43]. One of the most devastating pandemics in modern history, causing up to 100 million deaths worldwide, Spanish flu was also a significant cause of soldiers’ deaths back in 1918 [44]. Once again, proximity but also troop movement propelled the contagion to a new dimension; how the population dealt with this sanitary crisis and body gestion should encourage a complementary look at a more recent disaster. Epidemic deaths or deaths caused by infectious diseases were consequences of the war, and it is important to place them in the context of reflections on the lifestyles and survival of these soldiers. Typhus also devastated soldiers during WWII, greatly disabling the advancement of troops [45]. This information is important because the health statuses of soldiers necessarily influence the military strategies implemented. On this point, the example of the Crimean War (1853–1856) is emblematic, the cholera epidemic having greatly modified the operational plan initially envisaged.

Soldiers’ remains are important biological archives for better understanding the health context of this period at the turn of the 20th century. These bodies bear traces of both their past illnesses (such as the Russian flu) and their latest infections [46,47,48]. Here, again, technical and methodological advances in paleogenomics and paleoculture have made it possible to identify these traces of infection on human remains. In some cases, the cross-referencing of data from the study of bones and the results of pathogen identification allow a better understanding of the causes of death, such as the demonstration of a septic condition of a previously wounded soldier caused by gangrene [46]. In some cases, this pathological study can even be used to assess the health statuses of these young populations (to estimate the prevalence of certain diseases), such as the detection of cancerous lesions (e.g., Figure 6).

The genomic analysis of pathogens enables a better understanding of the evolution of their virulence and their mutation over time [49]. Therefore, being more than an indicator of population health, past pathogen analysis is a window to contemplating hypotheses on how actual pathogens could have evolved [50]. Climatic change, modern disasters, poverty and easy travelling make pathogen gestion an actual and relevant subject of interest. The opportunity for understanding pathogens’ evolutionary mechanisms according to past disasters can provide keys for epidemic prevention and addressing health issues.

## 4. Discussion

Mass mortality events, whether related to natural disasters or wars, past or present, bring with them the same expectations: to find the victims, identify them and return them to their families. This investigative work, this process of identification, necessarily calls for interdisciplinary research and requires the use of powerful tools to work with material (biological and artefactual) that is often very damaged and fragmented. Thus, progress in molecular techniques allowing the use of less and less biological material as well as knowledge about the human genome and epigenome are indispensable for overcoming the technical and material limitations that researchers face in the identification process.

The victims of the two World Wars are subject to a special legislative framework in France. The number of missing soldiers still present on French soil is considerable, and chance discoveries of their remains each year are very frequent. In fact, in France, the remains of these soldiers are legislatively considered special; they do not fall within the domain of archeology or forensics and are under the authority of the Office National des Anciens Combattants et des Victimes de Guerres (ONACVG) and the Ministry of the Armed Forces. Their discoveries, studies and management are strongly rooted in a duty of remembrance, both individual and collective [51].

The recent nature of the discoveries provides access to information that is difficult or impossible to use for other periods: administrative or private archives, photographic documentation, etc. Consequently, cross-analysis of these different sources makes it possible to identify their nationality and/or their regimental units and—sometimes, but in rare cases—to identify the fallen soldier [3,52,53].

The association of historical archives with bioarcheological data (particularly bone analysis) and the possibility of using cutting-edge molecular tools are essential for maximizing the chances of restoring to the dead of the two World Wars their identities and the honors that the nation owes them [3,54,55,56,57]. However, beyond this aspect of identification, there is another crucial aspect to consider in these contexts of mass mortality: the health context. Indeed, whether it was the cause of death or its consequence, the sanitary condition of these men was that of precarious populations often suffering from deficiencies and a lack of hygiene. Through the first years of their lives, these bodies carry their own health history (bacterial and viral infections, etc.). Knowledge of the pathogens linked to these contexts is therefore essential in order to have a diachronic vision of the years preceding the conflict and of the conflict period. Here, again, progress in paleogenomics makes it possible to search for small parts of pathogen genomes in ancient DNA matrices and to understand the causes of soldiers’ deaths from a new angle, the health angle, which is, at best, sometimes assumed but rarely demonstrated. The cross-referencing of data from the study of bone remains and the sequencing of human and pathogen genomes allows for a better understanding of the causes of death. Finally, studying the health of these soldier populations, often characterized by young individuals, also represents a real opportunity to understand the prevalence of certain diseases at that time and therefore offers new insight into current public health issues.

## 5. Conclusions

The French government, through the ONACVG, works with other organizations such as the Commonwealth War Graves Commission (CWGC) or the Volksbund Deutsche Kriegsgräberfürsorge (VDK) to share and improve the management of discoveries and exhumations. Each country must bury its dead, and the implementation of common administrative and methodological procedures makes it possible to build an indispensable international dynamic around the problem of missing soldiers and the duty to remember [42]. Completing the identification and individualization of soldiers will therefore necessitate calling upon all the archives that may be accessible, but also upon archeological data from excavations, the anthropological analysis of bone remains, and the exploitation of genetic and genealogical data in order to find the descendants or family links of soldiers for whom there is a possibility of identification. The different regulations, or simply the different customs and habits, in different countries, make this common international approach necessary.

However, beyond the intention to give an identity to these soldiers, most of whom remain missing, there is a major societal reality. How far is today’s society prepared to go (memorially, ethically and financially) to give an identity to these recovered bodies? In 2009, an interdisciplinary and international team excavated and identified the remains of 250 Australian soldiers who fell at Fromelles (Battle of the Somme, July 1916). A project large in scale, both technically and financially, was carried out over several months, including the genetic typing of thousands of Australians who had lost an ancestor during this battle in order to compare genetic profiles and enable the identification of the recovered soldiers [54]. Such willingness and investment are not systematic and can vary from country to country, depending on the historical and political context.

It is important to mention that biological profiling forms a solid base for comparison with the military archives, and social profiling can provide identities in several forms; none of them should be put aside. In contexts such as both World Wars, it is important to adapt the classic identification process as found in forensic anthropology. Methods categorized as secondary in Interpol DVI procedures are, here, necessary in the first place, to refine the research and tighten the archival research. The investigation of identity in both World Wars involves the fit of artefacts and biological profiles with archives, leading to DNA comparisons with family members if the identity is uncertain. We wish to promote a complementary approach with an interdisciplinary view.

Finally, while methodological and technical advances in recent years have allowed us to revise our approach to the study of human remains from the two World Wars, there is one inescapable truth. Time passes, sweeping away both the quantity and quality of the DNA of these missing victims, making it increasingly difficult to compare conclusive DNA profiles (between the dead and the living). As we have also seen, the interdisciplinary approach, offering a porosity between disciplines, still allows us to slightly push the temporal limits of the exploitation of certain clues by using methods at the cutting edge of molecular biology, such as in the identification process, but how far can we go?

Today, research on these populations of soldiers also allows us to gain insight into the health reality of these periods. These populations of the twentieth century faced two World Wars, and many major epidemic waves. The imprint left by these past humanitarian crises and their analysis are now a real public health issue. These remains constitute true biological archives on the major public health crises that occurred at the end of the 19th century and the beginning of the 20th century.

## Figures and Tables

**Figure 1 genes-13-01852-f001:**
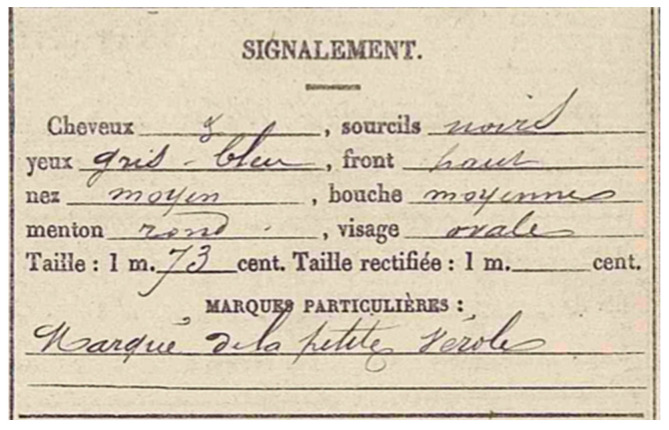
French register extract. Stature is mentioned as 1.73 m. **©** Departmental archives from Seine-et-Marne.

**Figure 2 genes-13-01852-f002:**
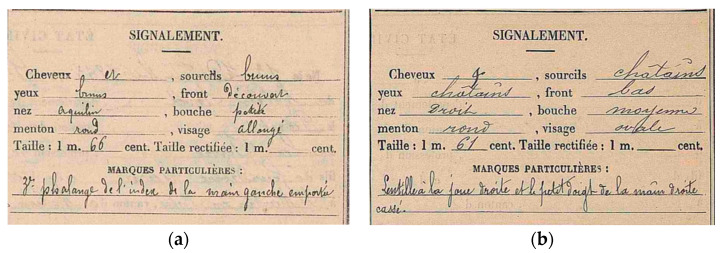
Military documents with mentions of particular conditions: (**a**) “3rd phalanx of the left hand amputated”; (**b**) “Broken right hand little finger”; (**c**) “Medium deviated spine”. **©** Departmental archives from Seine-et-Marne.

**Figure 3 genes-13-01852-f003:**
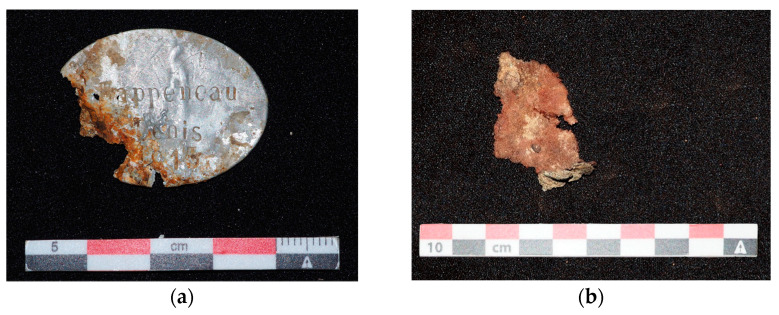
(**a**) Military ID tag; (**b**) Piece of red tissue from French 1914 pants. © Marine Meucci.

**Figure 4 genes-13-01852-f004:**
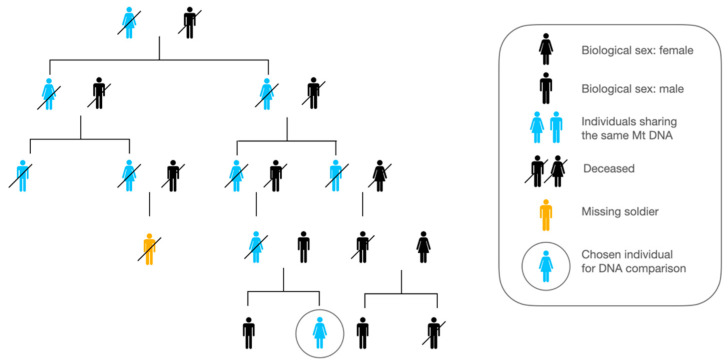
Example of how to choose and find the best family individual for DNA comparison. Genealogic representation.

**Figure 5 genes-13-01852-f005:**
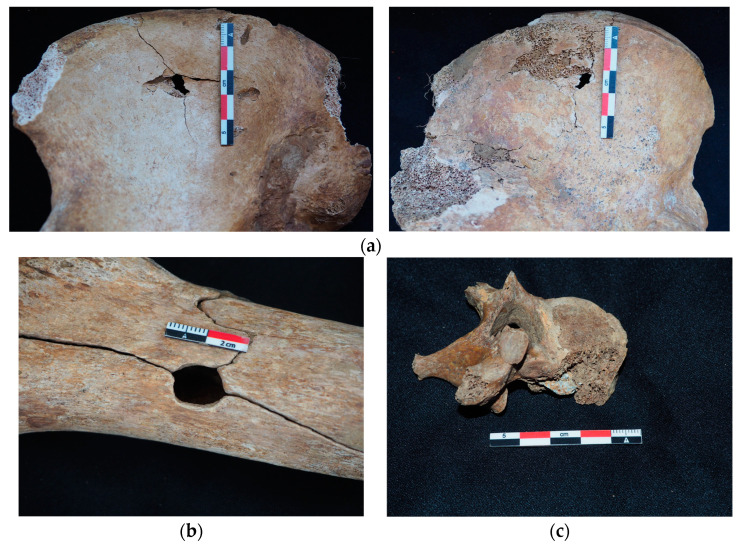
Bone trauma: (**a**) Entrance (left) and exit wound (right) from a ballistic trauma on the right coxal bone. (**b**) Peri-mortem fracture caused by a shrapnel ball on a femur anterior face. There was no exit wound. (**c**) Shrapnel ball stuck in the body of a thoracic vertebrae, lateral view. **©** Marine Meucci.

**Figure 6 genes-13-01852-f006:**
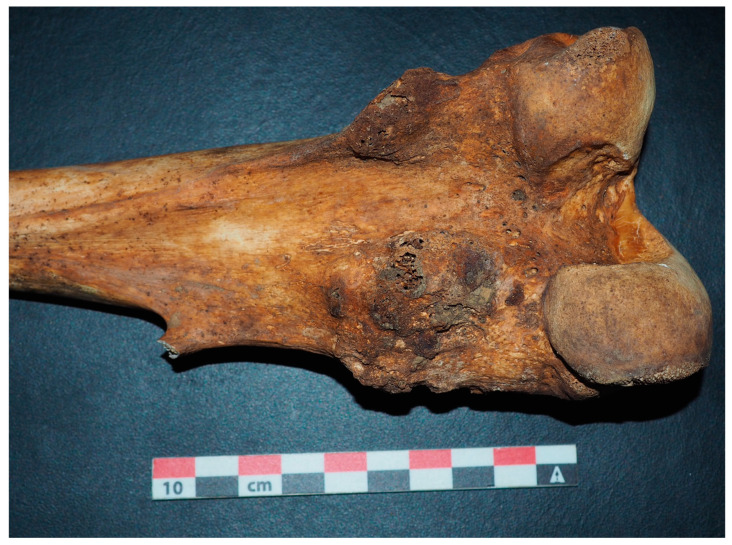
Distal femoral extremity with multiple osteochondromas, on a German soldier. © Marine Meucci.

**Table 1 genes-13-01852-t001:** Accuracy of sex estimation method using morphological and metric methodologies.

Bone	Methods	Accuracy
Pelvis	Morphometric DSP2; Bruzek et al., 2017 [6]	95%
Pelvis	Morphoscopic; Bruzek et al., 2002 [7]	95%

**Table 2 genes-13-01852-t002:** Methods used for age estimation and determining age range.

Bone	Methods	Age Range (Years)
Pelvis, auricular surface	Lovejoy et al., 1985 [13]	20–>60
Pelvis, auricular surface	Schmitt, 2005 [14]	20–>60
Pelvis, pubic symphysis	Schmitt, 2008 [15]	20–>60
All bones	Navega et al., 2022 [18]	19–101
All bones	Coqueugniot et al., 2010 [16]	7–29

**Table 3 genes-13-01852-t003:** Example of methods used on an age-known French soldier who died in 1915 at 20 years old and had been discovered in 2018.

Methods	Age Estimation
Lovejoy et al., 1985 [13]	20–29
Schmitt, 2005 [14]	20–29
Schmitt, 2008 [15]	20–29
Navega et al., 2022 [18]	18–35 with highest probability on 25
Coqueugniot et al., 2010 [16]	20–22 with highest probability on 21

**Table 4 genes-13-01852-t004:** Artefacts’ nature and the information they provide.

Artefacts	Material	Information
Helmet, shoes, bullet, military equipment	Various metals materials	Nationality
Money	Various metals materials	Temporality
Engraved objects	Various metals materials	Temporality, information about soldiers and/or identity
Pieces of uniform	Fabric/Metals	Rank/regiment
Military ID tag	Metal	Identity

## Data Availability

Not applicable.

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
