# Peer review of "The Skeletal Remains of Soldiers from the Two World Wars: Between Identification, Health Research and Memorial Issues"

_genes, 2022, doi:10.3390/genes13101852_

Round 1

Reviewer 1 Report

In this paper, the authors present the methods of identification used for the victims of the two World Wars in France. 

While the subject of the paper presents merit, I believe the manuscript presents major shortcomings which prevent its publication in its present state:

1. Extensive editing of English is required. The manuscript is not legible as is, many sentences do not make sense or should be completely reworded. Even some French expressions are written with mistakes (e.g. "Office National" and not "Nationale"). The degree of re-writing is so extensive that I would advice you to re-read and work on the manuscript in its entirety. 

2. The document reads as a book chapter and not a scientific paper: what was your sample size? Your methods? What were your anthropological results? On how many did you do DNA analysis for sexing? How many did you identify through DNA? Through other methods? The normal structure of scientific papers is not respected (Introduction, Materials and Methods, Results, Discussion, Conclusion). We do not understand what was the study and what results you obtained. As is, the manuscript is a general description of the identification process of war soldiers in France. How does it compare to other countries? What is different in this case compared to others? What do you recommend to forensic anthropologists? 

3. You seem to confuse the concepts of "forensic identification" and "biological profile". The biologica profile (i.e, the estimations of sex, age-at-death, population affinity, stature, as well as the analysis of diseases and trauma) does not aim at identification, but only serves to narrow down a list of potential candidates for identification, they are pieces of a larger puzzle if you will. While some features may act as factors of individualization (such as a prosthesis, pathological features, non-metric traits, etc) when compared to antemortem data, forensic identification is a strict process which only recognizes certain methods (e.g, INTERPOL DVI procedure).

4. Identification of the cause of death is not part of the role of the anthropologist. While some lesions may cause the death of the individual, the analysis is just that, speculative.

Reviewer 2 Report

Dear Editor of GENES and Authors,

I have read the paper titled «The skeletal remains of soldiers from the two World Wars: between identification, health research and memorial issues» with great pleasure, and I believe that it shows great scholarship and overall scientific quality and relevance. Actually, I have no suggestions to make regarding the scientific aspects of the paper. However, there are some imprecisions during the text that must be corrected, for example, in the Abstract: «After causing a mass disaster which decimate tens of thousands of soldiers…», substitute for «After causing a mass disaster which decimate MILLIONS of soldiers…». Also in the Abstract and elsewhere, the use of “don’t”, “isn’t”, etc, must be corrected as the use of contractions is not appropriate in scientific writing. Example: “…where they don’t belong…”, change “don’t” to “do not”. The writing of article will need to be carefully proofread and revised as there are many instances of less than desired quality of writing. Examples: «In France, death for both Great wars (1914-1918 and 1939-1945) are not recognized as forensic cases»; «To evaluate stature from soldier, the preferential method is based on long bones measurement [21].»; «In that way, soldiers died from violences…», or «Even if ballistic trauma is most common, traumas analysis in war context can remember the ones describes in forensic anthropology…»

My best regards,

Round 2

Reviewer 1 Report

This feels like a new manuscript, well-written, well-structured, with a clear aim and good descriptions. I appreciate the effort the authors put into the manuscript and can only commend them for their work.  I would only suggest one minor revision: I believe the sentence "Died for France" would be more accurate than "Death for France"